# Development of a Smart Helmet for Strategical BCI Applications

**DOI:** 10.3390/s19081867

**Published:** 2019-04-19

**Authors:** Li-Wei Ko, Yang Chang, Pei-Lun Wu, Heng-An Tzou, Sheng-Fu Chen, Shih-Chien Tang, Chia-Lung Yeh, Yun-Ju Chen

**Affiliations:** 1Institute of Bioinformatics and Systems Biology, National Chiao Tung University, Hsinchu 300, Taiwan; yanglala.bce04g@g2.nctu.edu.tw; 2Center for Intelligent Drug Systems and Smart Bio-devices (IDS2B), National Chiao Tung University, Hsinchu 300, Taiwan; wikibee.iie06g@g2.nctu.edu.tw (P.-L.W.); anantzou.06g@g2.nctu.edu.tw (H.-A.T.); 3Institute of Biomedical Engineering, National Chiao Tung University, Hsinchu 300, Taiwan; 4Institute of Biomedical Engineering and Nanomedicine, National Health Research Institutes, Miaoli 350, Taiwan; sanfo@nhri.org.tw; 5Information and Communications Research Division, National Chung-Shan Institute of Science and Technology, Taoyuan 325, Taiwan; tsjmark@yahoo.com.tw (S.-C.T.); austinyeh@msn.com (C.-L.Y.); graces801@gmail.com (Y.-J.C.)

**Keywords:** EEG, dry EEG electrodes, impedance, BCI, signal processing, helmet, SSVEP

## Abstract

Conducting electrophysiological measurements from human brain function provides a medium for sending commands and messages to the external world, as known as a brain–computer interface (BCI). In this study, we proposed a smart helmet which integrated the novel hygroscopic sponge electrodes and a combat helmet for BCI applications; with the smart helmet, soldiers can carry out extra tasks according to their intentions, i.e., through BCI techniques. There are several existing BCI methods which are distinct from each other; however, mutual issues exist regarding comfort and user acceptability when utilizing such BCI techniques in practical applications; one of the main challenges is the trade-off between using wet and dry electroencephalographic (EEG) electrodes. Recently, several dry EEG electrodes without the necessity of conductive gel have been developed for EEG data collection. Although the gel was claimed to be unnecessary, high contact impedance and low signal-to-noise ratio of dry EEG electrodes have turned out to be the main limitations. In this study, a smart helmet with novel hygroscopic sponge electrodes is developed and investigated for long-term usage of EEG data collection. The existing electrodes and EEG equipment regarding BCI applications were adopted to examine the proposed electrode. In the impedance test of a variety of electrodes, the sponge electrode showed performance averaging 118 kΩ, which was comparable with the best one among existing dry electrodes, which averaged 123 kΩ. The signals acquired from the sponge electrodes and the classic wet electrodes were analyzed with correlation analysis to study the effectiveness. The results indicated that the signals were similar to each other with an average correlation of 90.03% and 82.56% in two-second and ten-second temporal resolutions, respectively, and 97.18% in frequency responses. Furthermore, by applying the proposed differentiable power algorithm to the system, the average accuracy of 21 subjects can reach 91.11% in the steady-state visually evoked potential (SSVEP)-based BCI application regarding a simulated military mission. To sum up, the smart helmet is capable of assisting the soldiers to execute instructions with SSVEP-based BCI when their hands are not available and is a reliable piece of equipment for strategical applications.

## 1. Introduction

Conducting electroencephalographic (EEG) activity or other electrophysiological measures from human brain function might provide a nonmuscular medium for sending commands and messages to the external world, as known as a brain–computer interface (BCI). The purpose of BCIs is to convert human intentions into mechanical commands in order to control devices and machines without peripheral nervous systems and muscle movements. In recent years, a variety of BCI techniques have been developed and researchers have dedicated themselves to applying BCIs into real-world applications; for instance, motor imagery (MI) for stroke rehabilitation [1], steady-state visually evoked potential (SSVEP) for an alphabet speller [2], and P300 for controlling wheelchairs [3].

All aforementioned BCI methods exhibit several advantages and weaknesses compared with each other. For example, SSVEP shows a great signal-to-noise ratio [4] and relative immunity against artifacts [5], but it requires an external visual stimulus to trigger the human response. However, there remain several mutual limitations regarding applying such BCI techniques in practical scenarios; one of the limitations is the trade-off between wet and dry EEG electrodes. Wet electrodes provide better signal quality and lower impedance; nevertheless, they require skin preparation which removes the outer skin of the scalp before EEG acquisition due to the high impedance of the stratum corneum [6]. Moreover, the conductive solution is obligatory during the process, yet the improper application of conductive solution might lead to unsatisfactory measurements. Applying excessive conductive solution on the scalp would create electrolyte bridges between electrodes, causing them to interfere with each other [7]. In contrast, a lack of conductive solution would result in poor signal quality due to the drying out of solution [8]. Hence, it is inconvenient and time-consuming to administer wet electrodes in real-world applications.

In recent decades, the developments of dry electrodes have been springing up in the fields regarding EEG measurements and BCI applications [9,10]. Biopotential electrodes can be classified into wet Ag/AgCl, dry contact, and noncontact electrodes [11]. In this study, dry contact electrodes were brought into consideration and comparison because their characteristics are similar to our proposed hygroscopic sponge electrode: contact and ‘semi-dry’. Popular dry contact electrodes from recent years regarding EEG acquisition and BCI applications are described herein. The dry polymer foam electrode was developed by Lin et al. [12], which is a conductive polymer foam covered by conductive fabric and finally pasted on a copper (Cu) layer. For the purpose of conquering the issue of EEG acquisition posed by hair, a spiky structural design of electrode used. The spring-loaded electrode was manufactured by Liao et al. [13,14]. Each metal probe on the electrode was coated with gold (Au) to enhance conductivity without producing adverse effects on the skin. In addition, the probes were designed to include springs inside to avoid causing pain. In [15], foam-based and spring-loaded electrodes were compared with wet electrodes to validate their performance, and the results indicated that these electrodes performed similarly to wet electrodes under a resting state and motion tasks. The flexible silicon-based polymer electrode was developed by Chen et al. [16,17]. The pin-shaped structure of the electrodes exhibits the feature of flexibility and is thus capable of contacting the skin closely. The capacitive electrode was manufactured by Oehler et al. [18], being the first study to apply a capacitive electrode; however, the results suggested that it is not comparable with wet electrodes. Above all, dry contact electrodes provide suitable wearing experiences, but the high interelectrode impedance compared with wet electrodes still induces marked increases in noise level [19].

In this study, we proposed a smart helmet, which comes with semi-dry electrodes called hygroscopic sponge electrodes; the skin-contacting part is composed of sponge, which is hygroscopic and soft for the purpose of satisfying conductivity and comfort during EEG acquisition. Several properties of the electrode regarding EEG acquisition and BCI applications were investigated through the study, including the impedance of the electrodes and the correlation of the signals acquired via the classic wet electrodes and the proposed sponge electrodes. This novel electrode exhibits the advantages of both wet and dry electrodes: it is convenient to use as a wireless EEG accessory, and the impedance is lower than that of dry electrodes, yielding better signal quality. 

## 2. Materials and Methods

### 2.1. Development of the Smart Helmet

During the training for and execution of military missions, soldiers are usually equipped with combat helmets to protect their head; meanwhile, their hands constantly need to hold equipment, such as a rifle or a walkie-talkie. In this study, we proposed a smart helmet which integrated the novel hygroscopic sponge electrodes and a commercially available combat helmet for SSVEP-based BCI applications; with the smart helmet, soldiers can carry out additional instructions according to their intentions, i.e., through BCI techniques. Figure 1a shows the appearance of the smart helmet, and Figure 1b illustrates the interior view of it. The sponge electrodes were attached to the helmet for EEG acquisition, and the circuit was located inside the helmet for signal preprocessing. The circuit features a two-layered structure: the upper layer is the core analog signal processing unit, and the lower layer is the digital signal processing unit. The purpose of the two-layered design is to make the circuit reconfigurable, which means it can be assembled with other circuits without designing duplicate parts. The flowchart of EEG acquisition and preprocessing is shown in Figure 2, including filters, an analog-to-digital converter, and a transmission module. 

### 2.2. Development of Hygroscopic Sponge Electrodes 

As described above, wet electrodes require skin preparation and conduction gel to reduce the skin–electrode interface impedance [20]. The design of our proposed hygroscopic sponge electrode is adapted from [21]. This electrode consists of two materials to complete the skin-to-machine conduction: the first part is the cylindrical sponge with a hygroscopic nature, and the second part is the snap button for connecting the foam with the machine; here, we used the Ni/Cu snap. The hygroscopic sponge electrode measures 22 (L) mm × 22 (W) mm × 25 (H) mm.

In order to achieve lower impedance, some specifications are required for the sponge, including high water absorption, large contact area, and low resistance of the sponge material itself. Compared with polyurethane (PU) foam, pulp sponge features good electrical conductivity. There are two reasons for this. First, pulp sponge is composed of cellulose, which is a biomaterial with a hygroscopic nature. In addition to the porosity of the sponge structure, the hydrophilic cellulose molecule possesses higher water retention capacity. Second, the flexibility of wood-pulp fibers is larger than that of PU, so the soft sponge part can fit body surfaces perfectly, even in the areas covered with hair. Moreover, the electrode was designed to contact the skin via the dipped sponge, supported by a foam pad in order to avoid water effluence.

### 2.3. Experimental Design for Impedance Test of a Variety of Electrodes

In order to validate the performance of the developed electrode, an experiment of the impedance test was administered; the following sections elaborate the details.

#### 2.3.1. Subjects

Ten healthy volunteer participants were recruited for the experiment (average age: 24.4 ± 1.4 years; age range: 22–27 years; one female and nine males). None of them had any history of neurological defects. The experiment was approved by the institutional review board (IRB) of National Chiao Tung University, Hsinchu, Taiwan.

#### 2.3.2. Experimental Paradigm of Impedance Test

The experimental setup of the impedance test is shown in Figure 3. Three types of existing electrodes were adopted to be compared with the proposed hygroscopic electrode. From left to right in Figure 3a, the first one is the foam-based electrode referring to [12], and the second one is the classic wet patch electrode; this silver–silver chloride electrode is known to provide the most stable biological signal potentials [20]. It is assembled with an electrolytically coated silver disk and an adhesive foam pad filled with electrolytic gel [22,23]. The third one is a spring-loaded metal electrode from [13], and the last one is the proposed hygroscopic sponge electrode. Each of the four types of electrodes was attached to a wire, which was connected to the Neuroscan Synamps system (Compumetics Neuroscan) to acquire the magnitude of impedance through Neuroscan software, Scan 4.5 (Figure 3b). The foam-based electrode, the patch electrode, the hygroscopic sponge electrode, and the spring-loaded metal electrode were placed at Fp1, Fp2, O1, and O2 according to the international 10–20 system, respectively. The subjects were instructed to sit in a comfortable chair with their eyes open, staring at a fixed cross sign on the monitor. Meanwhile, the impedance measurements of four different electrodes were recorded. The measurements were taken down with an interval of one minute for a total of 20 min.

### 2.4. Experimental Design for Impedance Test of a Variety of Positions

#### 2.4.1. Subjects

Four healthy subjects participated in the experiment (average age: 23.3 ± 0.4 years; age range: 23–24 years; two females and two males). None of them had any history of neurological defects.

#### 2.4.2. Experimental Paradigm of Impedance Test

The impedance test experiment of a variety of positions was performed to validate the performance of the proposed electrodes. Eight sponge electrodes were attached to wireless EEG equipment, Mindo BR8, developed by the Brain Research Center of National Chiao Tung University, Hsinchu, Taiwan [24,25,26,27]. It is a wireless EEG cap that features dry sensors, miniature amplifiers, and a wireless telemetry, with eight fixed channels, namely Fp1, Fp2, Fz, C3, C4, Pz, O1, and O2 on it, according to the international 10–20 system. After setting up the equipment, the subjects were seated comfortably and maintained a relaxed posture in front of the computer. They were asked to stare at a fixed object, which was a cross sign on the monitor, during the recording period of 20 min. The impedance values were taken down for the duration of the experiment with an interval of one minute, the same as for the previous impedance test.

### 2.5. Experimental Design for Signal Validation

The EEG dynamics were acquired from sponge electrodes via Mindo BR8 and from disk electrodes (AgCl sintered) via Neuroscan simultaneously to validate the signal performance of the sponge electrode. In this experiment, eight channels of EEG signals were investigated, which acquired from one subject (age: 23, male).

The EEG activities were retrieved through the sponge electrodes that attached on the wireless Mindo BR8, and the channels we adopted were Fp1, Fp2, Fz, C3, C4, Pz, O1 and O2. The disk electrodes were placed beside the sponge electrodes to record the same region of EEG signal through Neuroscan (Figure 4a). The sampling rate of both devices were set up at 1000 Hz. The recording procedure is shown in Figure 4b. Subjects were asked to stare at the cross sign on the monitor for eyes-open recording; for eye-closed recording, the subjects were asked to close their eyes and relax. The total procedure lasted for 12 min; both eyes-open and eyes-closed recording sessions lasted for two minutes, and three trials of each were required.

### 2.6. Experimental Design for a Simulated Military Mission

After the impedance test and signal validation, another experiment would be implemented in order to validate the performance of the SSVEP-based BCI application utilizing the smart helmet. Furthermore, an algorithm called differentiable power was proposed to enhance the performance of the BCI system. 

#### 2.6.1. Subjects

In total, 21 healthy subjects participated in the experiment (average age: 23.1 ± 2.4 years; age range: 18–26 years; 15 females and 6 males). None of them had any history of neurological defects.

#### 2.6.2. Algorithm

One of the most common methods to detect a target, i.e., flickering the object being focused upon, is using a power spectral density analysis (PSDA) [28]. In recent years, there have been several new approaches coming out, such as the canonical correlation analysis (CCA) [29]. All of the algorithms attempt to derive the most possible target through various approaches. In this study, for the purpose of enhancing the performance of SSVEP-based BCIs, we proposed an algorithm based on PSDA, called differentiable power:
$$\mathrm{Differentiable}\;\mathrm{Power}=\frac{{P(f}_{T})}{\frac{1}{n-1}[\sum_{k=1}^{k=n}P\left(f_{n}\right){-P(f}_{T})]}=\frac{\mathrm{Target}\;\mathrm{power}}{\mathrm{average}[\mathrm{sum}\left(\mathrm{nontarget}\;\mathrm{power}\right)]},$$
where ${P(f}_{i})$ represents the power of frequency i, n is the amount of total targets, and T is the target. The algorithm was implemented in the system of the following BCI application.

#### 2.6.3. Experimental Paradigm of BCI Application

An established BCI application based on SSVEP was demonstrated to further strengthen the effectiveness of the proposed smart helmet. The framework was adapted from [21], which implemented a stimulated SSVEP-based military mission. Figure 5a shows the scene of the BCI application; the sponge electrodes were attached on the smart helmet for EEG acquisition. This application is composed of 30 trials, and each trial consists of three stages, including fixation, target search, and visual stimulus. Three soldiers would be the target, and there is a flickering icon in front of each of the soldiers, which flickers at an individual frequency. The system would select the target by deriving the EEG signal of the subjects and then shoot the derived target.

## 3. Results

In this study, four experiments were performed in order to validate the performance of the proposed smart helmet, including two impedance tests, a comparison between signals acquired from different electrodes, and an SSVEP-based BCI application. Besides the proposed smart helmet, two devices were adopted during the experiments. One was the well-known wet sensor EEG equipment: Neuroscan. The other was an eight-channel wireless EEG equipment called Mindo. The results of the experiments are elaborated as follows. 

### 3.1. Impedance Testing Results

Lower impedance indicates better conductivity between the scalp and the electrode; hence, two impedance tests regarding a variety of electrodes and scalp positions were investigated. Figure 6 shows the impedance display captured from Scan 4.5, Neuroscan. The color on each EEG channel indicates the magnitude of impedance. The colors sorted from high to low impedance were purple, red, brown, yellow, green, and blue, respectively (Figure 6a). Not only could the overview of the impedance be observed; the exact value of each channel could be obtained by clicking on a certain channel icon. For example, Figure 6b shows the impedance of channel Fp2, being 88 kΩ at that timepoint.

#### 3.1.1. Variety of Electrodes

Regarding the subjects, the impedance trends of all four electrodes is shown in Figure 7. The average values of resultant impedance at the point of the 20th minute were 413.2 ± 221.0, 148.1 ± 30.6, 123.5 ± 21.5, and 118.2 ± 13.8 kΩ for the spring-loaded metal electrode, the wet patch electrode, the foam-based electrode, and the hygroscopic sponge electrode, respectively.

Among all adopted electrodes, the spring-loaded metal electrode exhibited the highest impedance, and its impedance illustrated a fluctuating trend over time. Both the hygroscopic sponge electrode and the foam-based electrode exhibited lower impedance, and their impedance magnitudes showed a downward trend. Figure 8 separately plots the impedance comparison between the sponge electrode and the foam-based electrode. The impedance of the sponge electrode tended to stabilize after the moment of the ninth minute, according to the results of t-test between the magnitude of the 20th minute and the magnitude of each minute ahead (i.e., the magnitude of the 1st to 19th minute). Noted that the t-test was only administered on the impedance magnitudes of the sponge electrode. Although the sponge and the foam-based electrodes displayed lower impedance, the foam-based electrode was placed on Fp1, a location without presence of hair; the hygroscopic sponge electrode was placed on O1, which is a hairy side. Furthermore, except for the first minute, all standard deviations of impedance values regarding the sponge electrode were lower than for the foam-based electrode at every minute during the experiment. 

#### 3.1.2. Variety of Positions

Figure 9 shows the results of impedance acquired from four subjects at channels Fp1, Fp2, Fz, C3, C4, Pz, O1, and O2. The impedance acquired from these channels exhibited declining trends in the beginning, and then gradually became flat. The results suggested that the impedances at channels Fz, Fp2, Fp1, and Pz were capable of declining to a relatively low magnitude, around 120 kΩ; channels C3 and C4 demonstrated relatively high values, nearly 160 kΩ; and impedances at channels O1 and O2 showed the medium value of approximately 145 kΩ.

### 3.2. Signal Validation

In this part, we revealed the comparison between two signals, which were recorded from sponge electrodes via Mindo BR8 and disk electrodes (AgCl sintered) via the Neuroscan system. In Figure 10a–d, the O1, O2, C3, and C4 fluctuations of EEG potentials were plotted as the sample cases. The average correlation of O1, O2, C3, and C4 signals are 90.05%, 89.76%, 83.72%, and 92.07%, respectively. The values of correlation were obtained for a 0.2 s period, using a 0.1-s sliding window in two-second data. The same period and sliding window regarding correlation analysis were also applied to the 10-s data of these two EEG dynamics.

We also performed the correlation analysis to power spectral density (PSD). The EEG data was processed by fast Fourier transform (FFT) to get PSD; Figure 10e–h illustrates the PSD of channels O1, O2, C3, and C4 as the sample cases. As displayed, the average correlation of PSD acquired through these two different electrodes at channels O1, O2, C3, and C4 were 96.40%, 96.64%, 98.54%, and 98.90%, respectively. Table 1 separately illustrates the average correlation between two EEG signals recorded through the sponge and disk electrodes in two-second and ten-second EEG potentials, and also the PSD at channels Fp1, Fp2, Fz, C3, C4, Pz, O1, and O2. The average correlation of two-second and ten-second data were 90.03% and 82.56%, and the average correlation of PSD was 97.18%.

### 3.3. Performance of Simulated Military Mission

The proposed smart helmet was applied to the subject during the SSVEP-based BCI application regarding the simulated military mission. The application consists of 30 trials; Table 2 shows the number of hits (i.e., correct selection of targets derived by differentiable power algorithm) and the accuracies of all the subjects during these 30 trials in detail. By recruiting the proposed smart helmet with the attached hygroscopic sponge electrodes, the average accuracy of the SSVEP-based BCI application can reach 91.11 ± 7.58%.

## 4. Discussion

In this study, we developed a smart helmet and performed several experiments to investigate its characteristics and to validate its performance. The most important feature regarding the smart helmet is that it adopted the proposed hygroscopic electrodes. Impedance is considered as one of the most important properties regarding evaluating an electrode. Electrode impedance is expected to impact the signal-to-noise ratio during EEG recording [30]. In the traditional EEG recording system, the problem regarding impedance is typically solved by skin preparation, including cleansing and abrading the skin surface, and by applying conductive gel to the electrodes. Both of these solutions would cause some inconvenience to the examiners and subjects. In recent years, dry electrodes have been popular in the field of EEG recording and BCI because of their convenience. A variety of dry electrode solutions, such as structural and mechanical approaches, were investigated. The sponge electrode implemented such an approach through the structure and the nature of the material. Impedance tests and signal validation were performed to evaluate the proposed sponge electrode.

In the impedance test of a variety of electrodes, four kinds of electrodes were compared. However, the measurements of the electrodes were not bases on the exact same criteria because of the characteristics of each electrode. Two sides of the scalp were considered to administer the test; on the frontal side, the foam-based and the wet patch electrodes were placed because both these electrodes are not appropriate for application to hairy sites; on the occipital side, the spring-loaded metal electrode and the proposed hygroscopic electrode were placed to record the EEG dynamics. Another advantage of choosing frontal and occipital sides is that when performing EEG acquisition, all four electrodes were vertical to the ground, which minimized the effect of pressure caused by gravity. The results suggested that the impedance of the proposed electrode was comparable with the foam-based electrode, which exhibited the lowest impedance among three existing electrodes. Furthermore, unlike the foam-based electrode (Fp1), the proposed sponge electrode can be placed on sites with hair (O1).

The impedance test of a variety of positions revealed the influence of different channels regarding EEG acquisition. Due to the natural variabilities of humans and their skin conditions, such as skin thickness, the degree of skin hydration, and the number of hair follicles [31,32], the impedance of each channel varied. The channels with better performance (i.e., lower impedance) were Fz, Fp2, Fp1, and Pz; it was reasonable that Fp1 and Fp2 exhibited lower impedance because the absence of hair would allow the electrodes to contact the skin directly. The other channels with better performance were Fz and Pz, which were located on the superior side of the scalp. Because of gravity, the electrodes placed on the superior side of the scalp were expected to apply higher pressure on the skin, which would yield lower impedance as a matter of course [33,34]. With higher pressure applied on the scalp, the electrode would receive closer contact to the skin, a wider area of electrode–skin interface, and better stability to avoid the noise caused by motion or the movement of the electrodes [35]. The scalp–electrode impedance is approximately 120 kΩ, regarding the sponge electrode; although the impedance is relatively higher than the classic wet electrode, which normally measures around 20 kΩ, in a circuit design with an amplifier that implements around 200 MΩ input impedance, however, scalp–electrode impedance up to 200 kΩ is allowed for accurate signal acquisition with about 0.1% error [36]. As for further investigation of the effectiveness, signals acquired through sponge electrodes and disk electrodes were compared using correlation analysis. The results suggested that the signals were similar to each other; the minimum correlation of a total of eight channels exceeded 83% in two-second EEG potentials, exceeded 78% in ten-second EEG potentials, and exceeded 94% in power spectral density. Last but not least, the proposed smart helmet was applied to the BCI application of a simulated military mission. The subjects shot the target according to the result derived by the differentiable power algorithm in this SSVEP-based BCI experiment. In order to achieve higher accuracy of such SSVEP-based BCI applications, both low scalp–electrode impedances and an appropriate algorithm are obligatory. The previous experiments have evaluated the performances of the proposed hygroscopic sponge electrode, and the differentiable power algorithm was validated by the SSVEP-based BCI application. The average accuracy of the 21 subjects was higher than 90%, which indicates that the combination of the sponge electrodes and the differentiable power algorithm is adequate for SSVEP-based BCI applications.

## 5. Conclusions

The smart helmet with the attached hygroscopic sponge electrodes, which feature convenience and low impedance, was developed for EEG acquisition, especially for BCI applications. It possesses the characteristics of comfort, good accessibility, and high fidelity of signals because of the design, the materials, and the structure. Comparing the sponge electrode with existing dry contact electrodes, it is not only applicable for hairy sites on the scalp, but also exhibits lower impedance, which enhances the feasibility and performance regarding BCI applications. It also strengthens the effectiveness of the smart helmet with the low skin–electrode interface impedance and high correlation between the signals acquired from the proposed electrode and the classic Ag/AgCl electrode. To sum up, the smart helmet is capable of assisting soldiers to execute extra tasks with BCI techniques and is a reliable piece of equipment for strategical applications. 

## Figures and Tables

**Figure 1 sensors-19-01867-f001:**
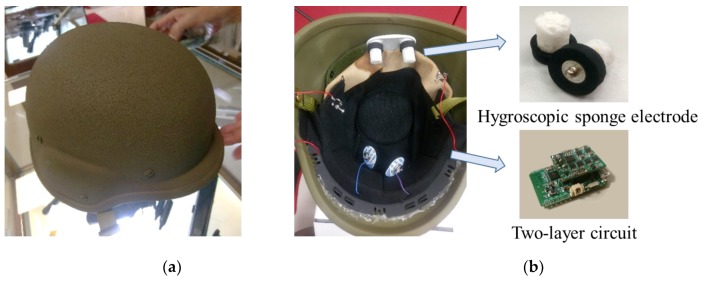
The appearance of the smart helmet. (**a**) The top view and (**b**) the interior view of the smart helmet; the sponge electrodes and the two-layered circuit are inside it. The smart helmet looks the same as an ordinary combat helmet from the top; however, it consists of novel electroencephalographic (EEG) electrodes and an internal two-layer signal processing circuit enabling it to acquire EEG signals and to perform brain–computer interface (BCI) applications.

**Figure 2 sensors-19-01867-f002:**
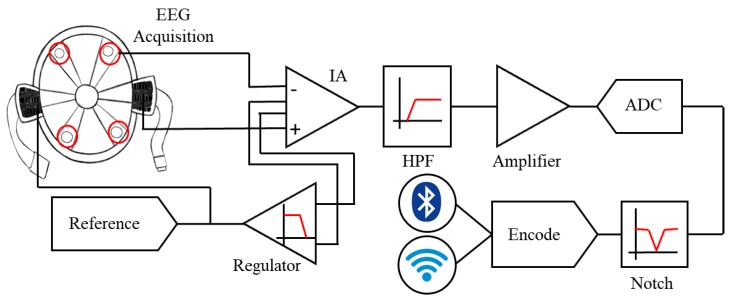
The flowchart of signal preprocessing. The EEG signals are acquired through the hygroscopic sponge electrodes, processed in the circuit, and then transmitted to the back-end for further analysis. In this study, the signal was analyzed and applied to a BCI application. Note: IA = Instrumental Amplifier, HPF = High-Pass Filter and ADC = Analog-to-Digital Converter.

**Figure 3 sensors-19-01867-f003:**
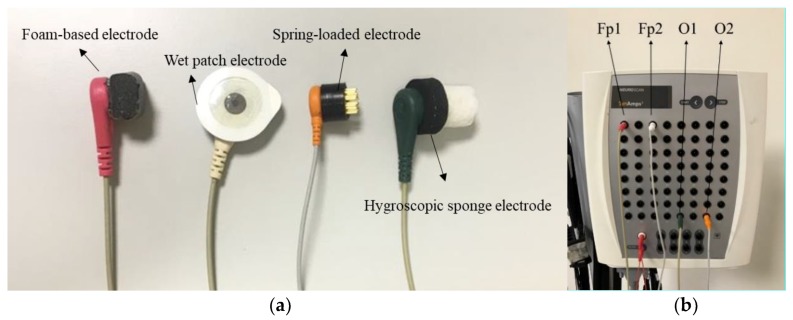
Equipment setup of the impedance test of electrodes. (**a**) Four kinds of electrodes were attached to the wires of Neuroscan amplifier to record the magnitudes of impedance. From left to right, the electrodes are the foam-based, wet patch, spring-loaded metal, and the proposed sponge electrode. (**b**) The wires attached with the electrodes were connected to the amplifier.

**Figure 4 sensors-19-01867-f004:**
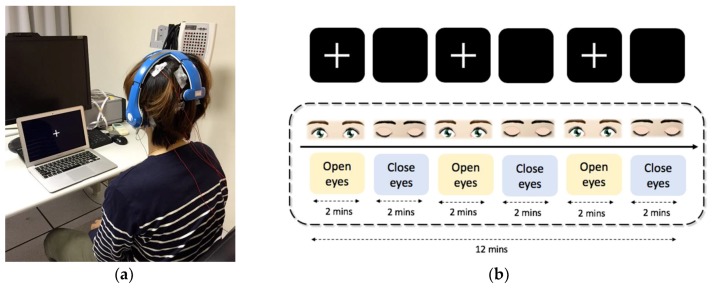
Experimental paradigm of signal validation. EEG signals of eight channels were acquired from Neuroscan (wet electrode) and Mindo (sponge electrode) simultaneously. (**a**) Experimental scenario; (**b**) experimental paradigm. The signals acquired from these two kinds of electrodes would be compared to validate the sponge electrode.

**Figure 5 sensors-19-01867-f005:**
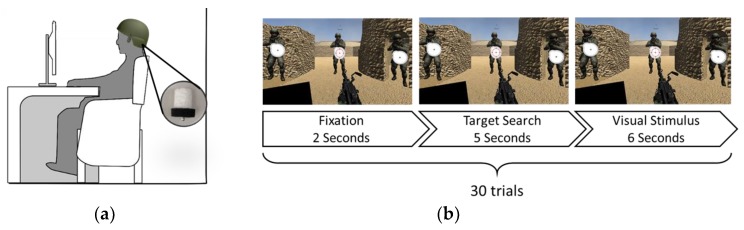
The framework of the BCI application. (**a**) The subjects wear the smart helmet containing the sponge electrodes for EEG signal acquisition and sit in front of a high-frequency monitor to perform the SSVEP-based BCI application; (**b**) The paradigm of the application. The application consists of 30 trials, which in total takes approximately 6.5 min.

**Figure 6 sensors-19-01867-f006:**
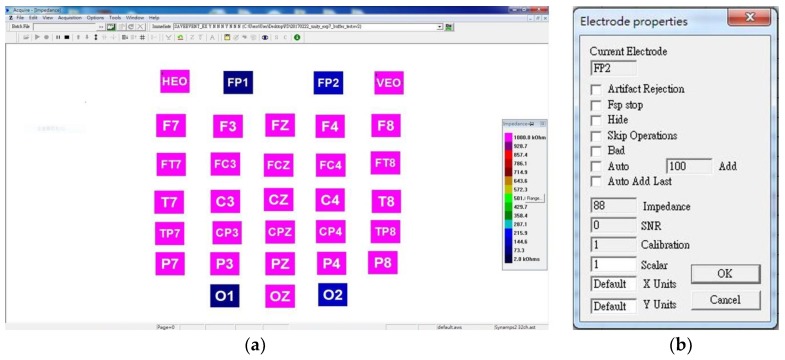
Impedance display of Neuroscan. (**a**) The impedance of the total scalp map represented in different colors; (**b**) example of the exact value of the impedance of each channel. This software used for the display of impedance is Scan 4.5, provided by Compumetics Neuroscan.

**Figure 7 sensors-19-01867-f007:**
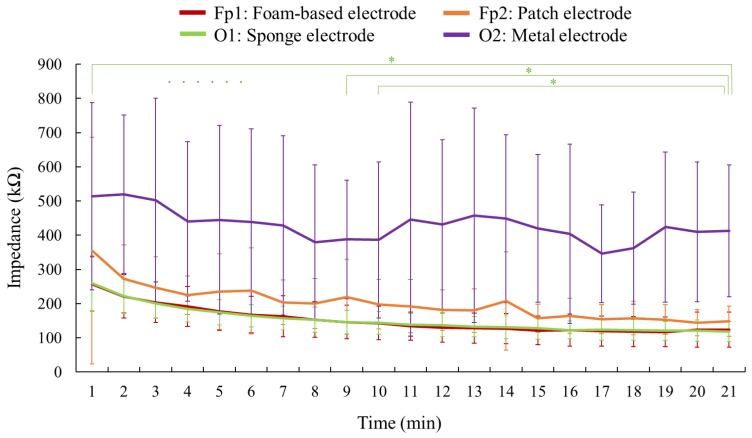
The impedance comparison of a variety of electrodes. This figure interprets the impedance trend of four kinds of electrodes in the twenty-minute experiment. The *p*-value of the t-test shows any significant differences, with *p* < 0.05 represented by an asterisk (*). The impedances of the sponge electrode (green line) began to stabilize after the ninth minute. The error bars indicate the standard deviation regarding the measurement of each minute.

**Figure 8 sensors-19-01867-f008:**
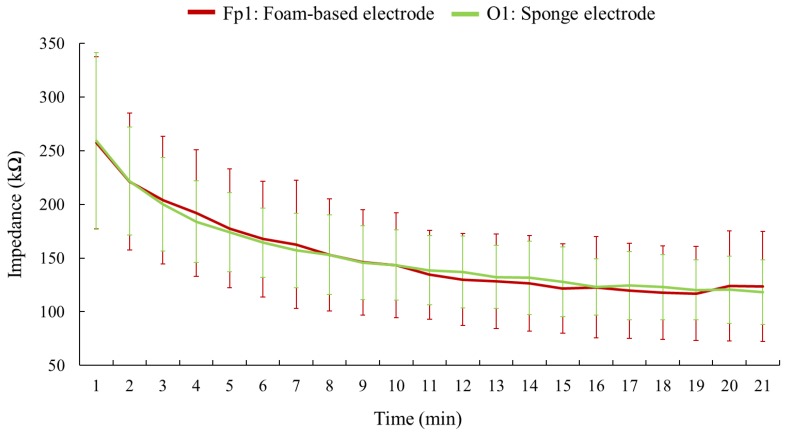
The impedance comparison between the hygroscopic sponge electrode and the foam-based electrode. This figure is a partial enlargement of Figure 7 in order to clarify the impedance values acquired from the foam-based electrode and the sponge electrode, which are quite close to each other. The error bars indicate the standard deviation regarding the measurement of each minute.

**Figure 9 sensors-19-01867-f009:**
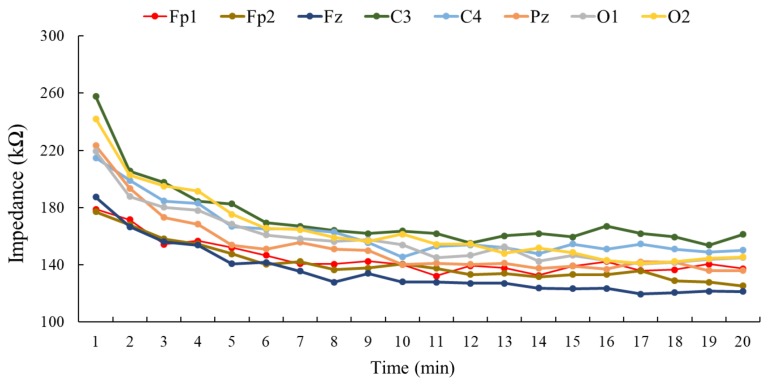
The impedance comparison of a variety of positions. This figure interprets the impedance trend of eight channels in the twenty-minute experiment. In the last minute, Fz, Fp2, Pz, and Fp1 exhibited lower impedance.

**Figure 10 sensors-19-01867-f010:**
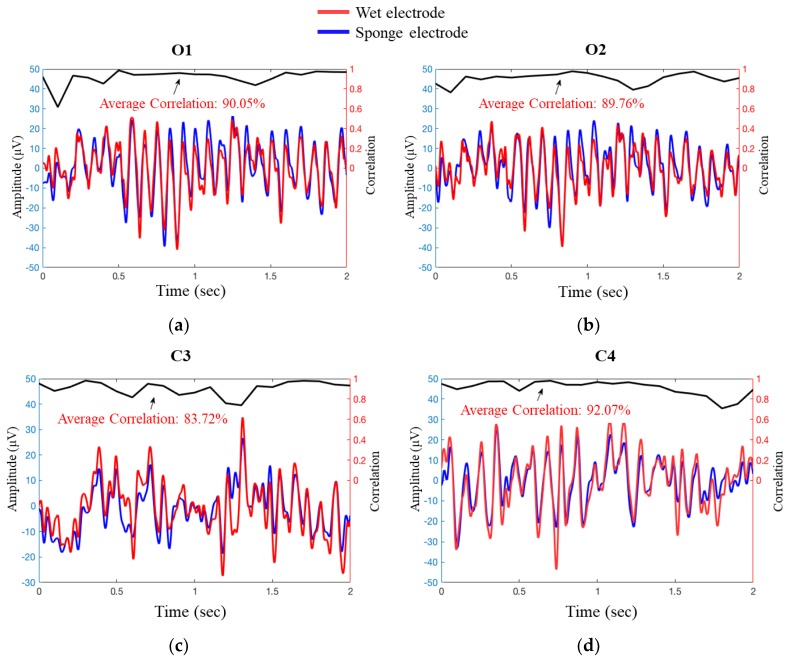
Signal validation between wet and sponge electrodes. (**a**–**d**) show the signal acquired from O1, O2, C3, and C4 channels in the time domain, along with their correlation. The EEG potentials recorded through the sponge electrode are shown by the blue line, the red line indicates the signal recorded through the disk electrodes, and the black line on the top demonstrates their correlation. Averaging was performed over a 0.2-s period using a 0.1-s sliding window; (**e**–**h**) show the signal of O1, O2, C3, and C4 channels, respectively, along with fast Fourier transform (FFT) and correlation. The blue line and red line represent the signal recorded through the sponge electrode and the disk electrode, respectively.

**Table 1 sensors-19-01867-t001:** Correlation of signals acquired from the wet electrode and sponge electrode.

Channel	Time Series		Power Spectral Density
	2 s	10 s	
Fp1	93.92%	87.99%	98.14%
Fp2	94.46%	87.27%	94.45%
Fz	86.84%	82.79%	98.57%
C3	83.72%	80.52%	98.54%
C4	92.07%	81.89%	98.90%
Pz	88.98%	80.38%	97.76%
O1	90.50%	78.70%	96.40%
O2	89.76%	80.91%	94.64%
Average	90.03%	82.56%	97.18%

**Table 2 sensors-19-01867-t002:** BCI performance of the simulated military mission using the hygroscopic sponge electrodes.

Subject No.	Hits	Accuracy (%)	Subject No.	Hits	Accuracy (%)
1	28	93.33	12	25	83.33
2	28	93.33	13	29	96.67
3	26	86.67	14	23	76.67
4	30	100.00	15	24	80.00
5	30	100.00	16	30	100.00
6	30	100.00	17	30	100.00
7	25	83.33	18	30	100.00
8	28	93.33	19	28	93.33
9	24	80.00	20	25	83.33
10	27	90.00	21	26	86.67
11	28	93.33	Average	27.3 ± 2.27	91.11 ± 7.58

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
