# Peer review of "Development of a Smart Helmet for Strategical BCI Applications"

_sensors, 2019, doi:10.3390/s19081867_

Round 1
Reviewer 1 Report
The authors have addressed my previous concerns. I recommend acceptance.
Reviewer 2 Report
The author has made enough corrections for previous comments. I have no other comments. The manuscript seems to be released in the current form.
This manuscript is a resubmission of an earlier submission. The following is a list of the peer review reports and author responses from that submission.
Round 1
Reviewer 1 Report
The manuscript reports a comparison between different types of electrodes. Although the topic is very interesting, the study presents several limitations and issues to be addressed before its publication on a journal paper.
Detailed comments are reported below:
- The Authors stated: "the best one within existing electrodes that averaged 123 kΩ referred to the classic wet electrodes?". This sentences is not correct since the classic wet electrodes (if well placed) usually reach an averaged impedence value of 20-30 (KΩ), considering central, parietal and occipital brain locations.
- How can the proposed hygroscopic electrodes be shaped in hairy areas, especially with curly hair? Please, describe it.
- There is not investigation about how long the different type of electrodes can keep low impedence values. This should improve the study.
- In order to better investigate and state the quality and reliability of the different types of electrodes, the Authors should include the wet electrodes (not the wet patch) among the tested ones, and use it as "gold-standard".
- The sample group size (n=4) is not enough for representative results on a journal paper. I would suggest the Authors to enrich the number of participants.
- From Figure 5, the electrodes, especially the occipital sites (i.e. O1, O2), seem quite far each other. Have the Authors measured the distances to quantify the eventual errors between the recorded and compared brain locations?
- t-test: With 4 participants the evidences would not be very robust. Also, the corresponding data distribution would be hardly normal, therefore a parametric statistical test is not appropriate. Furthermore, the statistical significance level (alpha=0.05) should be modified for multiple comparison depending on the number of comparisons.
- Figure 7: The impedence of the different electrodes should be reported for the same brain locations, otherwise it would not be possible to understand the eventual differences. Same comment for the averaged values at the different time "minutes".
- Figure 8: It is not possible to understand if the trends correspond to the averaged values, across the different kinds of electrodes, or only for one specific technology. Please, report it.
- Please, describe in detail how the correlation analysis have been performed (e.g. over time, over frequency bins), and if they were stastically significant.
- English check is recommended.
Reviewer 2 Report
The article gives an introduction to novel hygroscopic sponge electrode for EEG acquisition. The overall structure of the article is clear while the following problems still exist:
1. Delivery
There is faulty wording in several sentences of the article. For example, the sentence ‘ The other one was an eight-channel’ in line 163. In addition, many sentences do not read smoothly. Please check the article carefully.
2. Figures
Please replace the picture in Figure 1, 6 with a clearer picture. Besides, The picture in Figure 7 is out of shape. The picture has too many overlapping lines in to be recognized clearly. In addition, the font size of the scale of the axis in Figure 9 is too small. Please deliberate on the form of the representation.
3. Experimental Settings
In section 3.2, the writer discusses the similarity between the signal of wet electrode and sponge electrode. Other types of electrode should also be included (as presented before) to further strengthen the reliability of the results of the verification of equipment efficacy.
4. Some closely related works are missing, such as "ASCERTAIN: Emotion and Personality Recognition Using Commercial Sensors, 2018", "Predicting Personalized Image Emotion Perceptions in Social Networks, 2018", "Personality-Aware Personalized Emotion Recognition from Physiological Signals, 2018".
5. The part of the introduction to the equipment is too short and insufficient.